# How, and under what contexts, do academic–practice partnerships collaborate to implement healthcare improvement education into preregistration nursing curriculums: a realist review protocol

Lorraine Armstrong  ,[1] Chris Moir,[2] Peta Taylor[3]

¹Sport and Health Sciences, University of Stirling, Stirling, UK
²Centre for Post Graduate Nursing Studies, University of Otago, Dunedin, New Zealand
³Department of Health Practice, Ara Institute of Canterbury Ltd, Christchurch, New Zealand

**Correspondence to**
Lorraine Armstrong;
lorraine.armstrong@stir.ac.uk

## ABSTRACT

**Introduction** Internationally, healthcare improvement remains a clinical and educational priority. Consensus in Europe, Canada and the USA to implement quality improvement (QI) education into preregistration nursing curricula ensures students become equipped with the skills and knowledge required to improve practice. Now, New Zealand and Australia are beginning to implement QI education into their nursing curricula. However, QI education is complex; comprising multiple components, each influenced by the contexts under which they are developed and implemented. Evaluation studies of QI education unanimously acknowledge that academic and practice partnerships (APPs) are essential to optimally embed QI into preregistration curricula, yet it is not understood how, and under what contexts, APPs collaborate to achieve this.

**Methods and analysis** A realist review to determine how, and under what contexts, APPs collaborate to implement QI education in pre-registration nursing will be conducted using the Realist and Meta-narrative Evidence Syntheses: Evolving Standards Guidelines. International stakeholders will be consulted at each stage which includes (1) clarifying the scope of the review through empirical literature and tacit expert knowledge, (2) searching for evidence in healthcare and social science databases/grey literature, (3) appraising studies using the Evidence for Policy and Practice Information and Co-ordinating Centre weight of evidence framework and extracting data using Standards for QUality Improvement Reporting Excellence in Education Publication Guidelines, (4) synthesising evidence and drawing conclusions through the creation of context, mechanism and outcome configurations and (5) disseminating findings through conferences and peer-reviewed publications.

**Ethics and dissemination** Ethical approval was not required for this study. Findings will be disseminated to international nurse educators, leaders and front-line staff implementing QI education within their own academic and practice contexts through conferences and peer-reviewed publications.

**PROSPERO registration number** CRD42021282424.

### STRENGTHS AND LIMITATIONS OF THIS STUDY

⇒ Realist review is novel to explore quality improvement (QI) education in preregistration nursing.
⇒ Understanding academic–practice partnerships in developing QI education is novel.
⇒ Focus on the influence of context on QI education builds on the current literature.
⇒ A limitation of the study is that only English articles will be included.

## INTRODUCTION
### Healthcare improvement in nurse education

This millennium has witnessed a surge in healthcare improvement practice and educational development in response to practice errors and increased risk to patients' safety.[1] Therefore, developing workforce capacity and capability in improvement knowledge, skills and behaviours remains a priority on policy and educational agendas internationally.[2–6] In the past decade, countries such as the UK, Sweden, Spain, Canada and USA have developed consensus that quality improvement (QI) education in preregistration nursing curricula should be a regular and ongoing process, if students are to contribute to and lead practice improvements.[4 6–9] Recently, New Zealand has begun to embark on QI curricular integration, having established from 49 educational institutions that few display the foundations of quality and safety education.[10] It is recognised that failure to address this gap compromises graduate nurses' abilities to implement and sustain healthcare improvement. A similar drive for QI education in preregistration nursing curricula has also emerged from Australia where much of the focus is around partnership working.[11] The

authors of this review protocol have integrated, or are about to integrate, QI curricula into their nurse education programmes in the UK and New Zealand and have a shared purpose to develop valuable insights into how academic and practice partnerships (APPs) collaborate to optimally embed QI into nurse education.

To date, QI educational content in preregistration nursing has included improvement methods, quality indicator measures, the Model for Improvement, Plan-Do-Study-Act cycles, root cause analysis, systems thinking, interprofessional learning, clinical governance, data, human factors and evidence-based practice, which, when delivered through an experiential learning pedagogical approach, is regarded most appropriate to equip preregistration students with the necessary skills and knowledge about QI. Supplementary approaches include didactic learning, seminars, group work, self-directed learning and/or simulation.[12]

QI education is regarded as a complex social phenomenon, whereby its numerous components such as the learner, the curricula, the faculty and the practice setting all interact on multiple levels of the system. They are also subject to the varying contexts under which they are developed and implemented.[13] Considering QI education from a 'systems thinking' perspective is useful here, because this approach views the components as a connected whole rather than separate component parts.[14]

There are many component parts which contribute to the complex nature of developing and implementing preregistration QI nurse education. The literature describes these as: developing the curricula, delivering teaching content, creating real life practice-based QI opportunities, developing staff expertise in improvement methodologies, securing mentorship and support, bridging support through information provision and link staff, updating student mentors on QI education through workshops, developing resource handbooks, seeking approval of students' improvement ideas, access to data, governance of ethical student practice, creating a QI culture where practice staff facilitate and mediate between students and other staff members, ongoing development and evaluation of QI education, and lastly supporting dissemination activities to showcase students QI work.[14–18] Taking account of these multiple components, it is evident that developing and implementing complex QI educational curricula requires synchronised planning between academics and practitioners, if students are to engage in meaningful practice-based improvement opportunities.

Despite a growing interest emerging from academics and practitioners to implement QI nurse education, there is currently no research to inform how, and under what contexts, APPs collaborate to achieve this.[10 11 19]

### Academic–practice partnerships

Academic–practice partnerships (APPs) are defined as 'strategic associations between educational and clinical facilities that are established to advance their mutual interests in nursing practice, education and research'.[20] Their purpose ranges from developing preceptorship education to clinical nurse leadership initiatives in hospitals, community health and public health agencies, nursing homes, schools and governmental agencies.[20 21] They operate on strategic, tactical, operational, interpersonal and cultural levels.[22]

There has been a recent review that considered the effectiveness of nurse APPs on costs, employability, work-readiness, confidence, competence and stakeholder satisfaction, but not in developing or implementing nurse education.[23] The same review also encourages future research to focus on how APPs function and sustain under different contexts. As such, how APPs function and sustain under different contexts when developing and implementing QI nurse education remains our focus.

We do know though that nursing APPs have been investigated in the management literature which attribute several key principles to successful collaboration.[22] These include creating a shared vision, establishing accountability, nurturing appropriate people engagement with balanced relationships, maintaining trust and respect, practising good communication and putting governance in place. Through these principles, APPs benefit from improved organisational efficiencies, enhanced opportunities for innovation and enhanced recruitment and retention.[22]

Similarly, there is evidence from the management literature of investigations into how different types of partners in healthcare improvement practice and research, not education, generally work together.[24] It is suggested that healthcare improvement partners require three types of skills to function and be sustained. These relate to technical skills (Lean, PDSA and run charts), soft skills (communication, leadership, team skills) and learning skills (knowledge sharing, encouraging participation and collective learning). In the absence of one skill subset, it is reported that healthcare improvement partners experience difficulties leading change.[25]

Lastly, a scoping review collating evidence of the effectiveness of partnerships in clinical nurse education has also been undertaken; however, the eligible studies in this review adopt weak evaluation measures and use low-quality appraisal instruments to determine overall effectiveness.[23]

If nursing APPs are to develop and implement QI education under their own practice and education contexts, then insights into the experiences of this process is essential.

### What will a realist review add to the field?

Realist methodology is a theory-driven approach appropriate for investigating how APPs work together, because it not only considers the complex nature of QI education which operates as a social intervention but recognises the different contexts and mechanisms that contribute or not to the QI educational programme's success.[26] The Medical Research Council has recently updated their

complex intervention guidelines to encourage realist approaches for evaluation studies.[27]

A previous review that has investigated the effectiveness of APPs working in clinical education, and one examining the methodological rigour in QI curricula, both criticise studies for the research approaches they have adopted.[23 28] This is because, predominantly, researchers investigating QI education have asserted a positivist stance. Traditionally, positivists sit within the natural (physical) sciences, are associated with quantitative approaches and claim that an absolute truth of knowledge exists.[29]

Realist researchers also sit within the natural (physical) sciences as postpositivists but challenge an absolute truth theory in favour of a realist perspective. Both sides proclaim broadly similar effect and outcome orientations, but postpositivists argue that only an approximation of truth exists, and as such focus on the mechanistic factors which determine outcomes, as opposed to the outcomes themselves.[30]

Positivist stances and quantitative approaches are regarded as too simple to determine causal linkages between complex interventions and outcomes because they discount the all-important concept of context.[31] Ignoring context is considered to be the antithesis of healthcare improvement principles and is instead a prerequisite to understanding why success ensues in one instance and not in another. The context debate is infinitely evolving, and in-depth discussions are widely available.[24 25] However, in general terms it relates to everything. Context has become the central focus of QI nurse education studies recently, in which ethnographic methodology is used to explore the contexts that influence student nurses QI learning experiences in the practice setting.[32]

There have been many frameworks developed to study context in healthcare improvement.[33] For instance, The Model for Understanding Success in Quality improvement targets contexts relating to micro-systems; these include the QI team, QI support and capacity, the organisation and the external environment[34] while others consider context as the outer components (political, social and cultural) and inner components (organisational structure, size and performance).[35] In terms of realist reviews, Greenhalgh and Manzano[36] have more recently identified context as[1] tangible, fixed, observable features that trigger mechanisms or[2] relational and dynamic features that shape the mechanisms through which the intervention works.

For the purpose of exploring 'contexts', this review protocol will draw on Tess et al's conceptual framework of integrating quality and safety into medical education.[37] We did not come across a similar model which exists for nurse education in the literature. The concepts in their model will form our preliminary investigation into the contexts APPs might operate under. The contexts we are initially guided towards, but will not be restricted by, are organisational culture, academic–practice alignment, infrastructure, curricular resources, faculty-practice

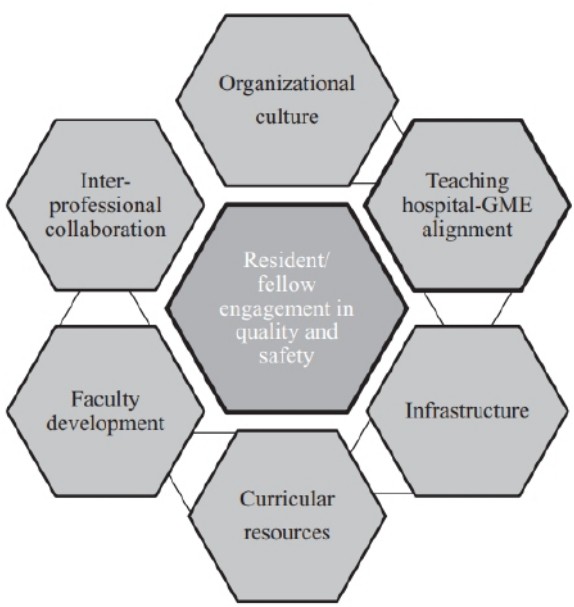

**Figure 1** Tess *et al*'s[37] (2015) framework for integrating QI education. QI, quality improvement.

development and interprofessional collaboration (see figure 1). As our review progresses and our data evolves, we will develop context themes iteratively and report on any further developments in the final write up of the study.

## METHODS AND ANALYSIS

For completeness and transparency, this review protocol will uphold the recommendations of Preferred Reporting Items for Systematic Reviews and Meta-Analysis Protocols, where it relates to our realist review design and is available in additional file one.[38] The synthesis process of this study protocol commenced in June 2020 and is predicted to conclude in June 2024.

### Patient and public involvement

Patients and public were not involved in the development of the realist review protocol.

### Study design

Our realist review of the literature will be guided and conducted in line with the Realist and Meta-narrative Evidence Syntheses: Evolving Standards (RAMSES) Guidelines.[39] This approach will enable a rigorous process to be followed ensuring that both methodological decisions remain appropriate to the review type and anticipated data set. The steps are (1) clarifying the scope of the review through empirical literature and the authors' tacit expert knowledge, (2) searching for evidence in healthcare and social science databases, and grey literature, (3) appraising studies using the Evidence for Policy and Practice Information and Co-ordinating Centre (EPPI) weight of evidence framework and extract data using SQUIRE-EDU (Standards for QUality Improvement Reporting Excellence in Education): Publication

Guidelines, (4) synthesising evidence and drawing conclusions through the creation of context, mechanism and (5) outcome (CMO) configurations and disseminating findings through conference presentations and peer-reviewed publications.

## STUDY METHODS
### Clarifying the scope
The original scope of our realist research stems from the findings of our previous research. First, an integrative review which explored QI education in preregistration healthcare examined QI educational interventions in preregistration healthcare. This study concluded that there was a need to develop and strengthen APPs to embed QI in nurse education.[12] Equally, a drive from New Zealand and Australian healthcare sectors to embed QI into their nursing programmes has broadened the scope of the review and created an opportunistic and timely collaboration to develop insights into how APPs have implemented QI education internationally[10 11 19] This evolving landscape, coupled by the authors' academic and clinical tacit knowledge of developing and implementing QI nurse education, has shaped the initial aim and objectives early in the process.

### Study aim
The aim of our realist review is to develop an understanding of 'how' and under what contexts APPs collaborate to implement healthcare improvement education into preregistration nursing curriculums.

### Study objectives
The key objectives of the research review are to:
1. Identify the contexts under which APPs collaborate to implement QI into preregistration nurse education.
2. Analyse the mechanisms and outcomes associated with each context under which APPs collaborate to implement QI into preregistration nurse education.
3. Explore relationships emerging across and between the contexts, mechanisms and outcomes in which APPs collaborate to implement QI into preregistration nurse education.
4. Explain the causal pathways of how APPs collaborate to implement QI into preregistration nurse education.
5. Disseminate findings so that nursing and academic institutions, new to QI education, are encouraged and well informed to develop their own highly effective APPs.

The next step entailed undertaking a broader exploration of the literature to enable us to refine the review's parameters. We identified a key difference between international countries; that being some healthcare systems such as the UK, New Zealand, Australia and Canada, were publicly funded compared with that of the USA which is privately funded. Exploring both healthcare systems was believed to be too vast for the team's time and resource, however, our scoping activity retrieved lower-than-expected numbers of studies from outside of the USA, who remain international leaders in healthcare improvement education and research. As a result, we broadened the review's international range and will contemplate whether this difference in healthcare systems could act as a potential 'context' category under which APPs collaborate.

Lastly, the core research team held a stakeholder engagement exercise to assist in the development of the study in December 2021. Stakeholders that were contacted included international academic and practice staff operating on multiple levels of the healthcare and academic organisations such as preregistration and postregistration nurses, nurse directors, nurse academics, clinical nurse educators, nursing directors of quality and nursing journal editors. We provided an opportunity to shape the realist review protocol, make adaptations to the study methods, discuss pressing issues and then actioned suggestions to refine the scope of the review. Stakeholders are listed in the acknowledgements. Consideration of the value that patients, carers and the public add to the project will be regularly reviewed by the team, acted on for input where applicable and reported in the final review manuscript.[40]

### Developing our programme theory
Programme theories in realist review relate to the intervention under investigation, in this case the QI educational programme. They seek to offer a theoretical explanation of how the intervention works, for whom and under what contexts. Seeking out assumptions of how interventions operate under different contexts, through a variety of mechanisms which produce different outcomes, is a continuous, collaborative and an iterative process which uses theoretical and empirical literature to populate and test initial 'rough theories'.[39] A number of initial rough theories about how APPs collaborate to implement healthcare improvement education have influenced our investigation:
► Experiential learning is the most appropriate pedagogical approach to equip student nurses with adequate QI knowledge and skills. Therefore, we anticipate that early formation of APPs is needed to streamline quality teaching and learning experiences.
► The international landscape varies, and some APPs are more experienced at implementing QI education than others. It is possible that policy drivers and/or professional standards from within each country will determine to a greater/lesser extent, the priority by which QI education is implemented.
► APPs operate on multiple levels of the organisation just as QI education comprises multi components that requires a 'systems thinking' perspective. We predict that multiple APPs may operate simultaneously, over multiple levels of the organisation to implement one QI education programme, rather than one APP operating on an individual basis.

In line with realist review methods, our 'initial rough theories' stated above, will continue to be refined and/or expanded in an iterative manner as we explore further literatures, and analyse and populate our data into our context framework.[39]

### Search for evidence

Approaches to evidence searching in realist reviews are diverse and dependent on the discipline and study objectives; on average 3–4 iterative phases are undertaken.[41] Our protocol remains at the earliest phase of our evidence search after which we will undertake further phases before completion.

Atypically, we employed traditional systematic searching for empirical literature early to complement our scoping review. While this method is typically classed as the 'main' search in realist reviews,[41] our familiarity with the dearth of existing research in this discipline, led us to plan our 'main' evidence search within the unpublished literature, internationally. Therefore, as an adjunct to our scoping review we undertook early empirical searching of QI education within social science and healthcare databases.

### Electronic databases

Electronic databases: Cumulative Index to Nursing and Allied Health Literature (CINAHL), EMcare, Medical Literature Analysis and Retrieval System Online (MEDLINE), Science Direct and Scopus were systematically searched. Studies were limited to peer-reviewed and English language studies. Aligning to the inception of QI nurse education in the millennium, databases were searched from 2000 to 2021. A primary search for MEDLINE was developed using MeSH (Medical Subject Headings) terms. Subsequent searches were translated to each database through Boolean terms, truncation, wildcards and proximity searching to maximise relevant papers. A primary sample search strategy from MEDLINE is presented (see box 1).

In the next phase, our 'main' evidence search will involve locating international policy documents, online blogs, professional body documentation, healthcare and educational establishments' websites, social media, dissertations, opinion articles, editorials and Google Scholar. We will gather evidence to support or refute our programme theories. Search terms will draw on the MEDLINE strategy as well as focusing on policy, management, governance, workforce, system thinking and healthcare education.

Finally, we will update previous search strategies and present evidence to our stakeholders for their comments. We will seek the opportunity to ask them to signpost us to potentially eligible evidence. Once saturation is deemed to be reached, results yielded from each phase will be transparent and illustrated in a flow diagram.[42]

### Reference list searching

As with current evidence, all future evidence will undergo scanning of reference lists to identify eligible evidence not detected initially. Where difficulty in retrieving evidence

---

**Box 1    Developed search strategy for MEDLINE using OVID**

1. Quality improvement/
2. Quality improvement.tw.
3. Quality Assurance, Health care/
4. Quality assurance.tw.
5. improvement science.tw.
6. improvement methodology*.tw.
7. Improvement model*.tw.
8. service improvement.tw.
9. CQI.tw.
10. plan do study act.tw.
11. PDSA.tw.
12. Plan do check act.tw.
13. PDCA.tw.
14. Total Quality Management/
15. Total quality management.tw.
16. TQM.tw.
17. 1 or 2 or 3 or 4 or 5 or 6 or 7 or 8 or 9 or 10 or 11 or 12 or 13 or 1 or 14 or 15 or 16
18. Interinstitutional Relations/
19. Interinstitutional relations*.tw.
20. Public-Private Sector Partnerships/
21. Public-Private partner*.tw.
22. academic practice partner*.tw.
23. intersectoral collaboration?.mp (mp=title, abstract, original title, name of substance word, subject, heading word, floating sub-heading word, keyword heading word, organism supplementary concept word, protocol supplementary concept word, rate disease supplementary concept word, unique identifier, synonyms)
24. collaborat*.tw.
25. communit* of practice.tw.
26. 18 or 19 or 20 or 21 or 22 or 23 or 24 or 25
27. Health Personnel/ed (Education)
28. Health personnel education.tw.
29. Health staff education.tw.
30. Exp Education, Nursing/
31. Nursing education.tw.
32. allied health education.tw.
33. or 28 or 29 or 30 or 31 or 32
34. 17 and 26 and 33
35. Limit 34 to (English language and yr="2000-Current")
36. From 35 keep
1, 12, 14, 28, 37, 43, 57, 66, 71, 74, 81, 94-96, 102-103, 117, 123, 128, 130-131, 136, 140, 144, 147, 149, 151, 163, 169, 174, 198, 219

---

occurs, authors or organisations will be contacted to request the evidence.

### Inclusion/exclusion criteria
#### Inclusion

Empirical searches, both current and future, will include international peer-reviewed studies of all designs where abstracts are written in English from 2000 to 2021 and, respectively, 2023 when searches are rerun. Studies will need to describe an element of APPs collaboration to implement QI education in preregistration nursing. We accept 'quality improvement education' where it describes the teaching, learning or utility of a healthcare

improvement model and details a theory and practice-based element.

### Exclusion

Current and future empirical evidence will exclude studies which make no reference to how the APPs collaborated, refer solely to online learning or patient safety content, or do not include preregistration nursing students.

### Data collection and management
#### Search for evidence

For all current and future evidence, reviewers (LA, CM and PT) will conduct and save searches within corresponding databases where applicable. To ensure each search remains accessible, reproducible and transparent a manual record keeping log will continue to be produced. To allow reviewers access, studies eligible for screening will be uploaded electronically to a shared named folder within Microsoft Teams. Duplicates will be removed.

#### Screening and selection of studies

Screening and selection of current and future evidence will follow the same process by which reviewers (LA, CM and PT) will work independently to screen titles and abstracts and/or executive summaries from retrieved evidence using predetermined inclusion and exclusion criteria. Ambiguities around papers will be discussed and the decision to include or exclude them will be made jointly. Evidence for inclusion will continue to be transferred to a separate shared named folder to allow reviewer access for data extraction. Some excluded evidence may remain valuable for later discussion of the review findings, as such these will be stored in a separate share folder in Microsoft Teams. To ensure consistency in the application of inclusion and exclusion criteria, one reviewer with subject expertise (LA) will review a 10% random sample of the evidence to confirm inter-rater reliability. Discrepancies will be resolved by discussion.

#### Data extraction and appraising the evidence

Reviewers (LA, CM and PT) will extract data with cognisance of SQUIRE-EDU: Publication Guidelines for Educational Improvement.[43] Reviewers will also focus on extracting data under the headings which relate to the key 'contexts' of interest from Tess *et al*'s framework.[37] These include the organisational culture, practice–academic alignment, infrastructure, curricular resources, faculty-practice development and interprofessional collaboration. Mechanisms relating to each 'context' will be extracted and included under each heading. Reviewers will also extract the outcomes of interest that are detailed by authors in each study. One reviewer with expert subject knowledge (LA) will compare a 10% random sample to ensure consistency. Discrepancies will be resolved by discussion or a fourth reviewer where necessary.

A realist review is predicted to retrieve heterogeneity in study design, therefore, using a singular hierarchical appraisal tool for exclusion will not be fit for purpose or add value to answering the study's questions. The EPPI

Weight of Evidence framework which considers the use of judgement in relevance and quality will be used (table 1). This framework allocates each study with a weight of high, medium or low in relation to three key areas: (a) trustworthiness of results of study, (b) appropriateness of study design to the review question and (c) appropriateness of focus to answering the review question. The results of (a), (b) and (c) are combined and given an overall weight. This method allows the worth of each study to be identified for the realist synthesis rather than being a preliminary prequalification exercise.[44] To ensure consistency in overall Weight of Evidence allocated, reviewers will meet to compare a 10% random sample of low, medium and high graded papers and discrepancies resolved through discussion or a fourth reviewer if necessary.

### Analysis and synthesis of evidence

As a general framework, the RAMSES guidelines will be applied.[39] Analysis in realist reviews is primarily concerned with refining theory-driven explanations, in this case, of how APPs work collaboratively to implement QI education under different contexts. Analysis will involve, first: identifying contexts under which APPs collaborate and informed but not limited by a preliminary framework of QI education developed by Tess *et al*.[37] We will discuss new emerging contexts and consider what supporting evidence warrants its inclusion to refine our programme theories. In line with systems thinking, we will explore the relationships that exist as a collective whole, in order to create a broader perspective of QI education as a holistic system and add depth to our understandings.

Next, the mechanisms and outcomes associated with each context will be investigated and listed as CMO configurations. A CMO configuration 'is a statement, diagram or drawing that spells out the relationship between particular features of context, particular mechanisms and particular outcomes'.[39] The causal relationships within each CMO configuration will be explored and listed. Recurring themes will be identified and grouped as demiregularities (semipredictable patterns). We will consider how the evidence supports or disproves our initial rough theories of how APPs collaborate to implement QI education into preregistration nurse education. The findings will be presented as a narrative synthesis which the contexts and any relationships between them will be discussed. The demiregularities associated with each context will be embedded into this discussion as we see appropriate, and our conclusions will take account of the current literature landscape to make future recommendations.

### Disseminate, implement and evaluate

The research will be disseminated through published works, local and international conference presentations. The findings will be used by two New Zealand tertiary institutions and their partnering district health boards to develop effective APPs to implement QI education in their nursing programmes. A realist process evaluation will be undertaken to advance the

**Table 1** EPPI weight of evidence

| | WoE A | WoE B | WoE C | Overall WoE |
|---|---|---|---|---|
| **Weight of evidence** | **Trustworthiness of results of study** | **Appropriateness of study design to review question** | **Relevance of study focus to answer review question** | **A, B and C Combined** |
| | High:<br>1. Intervention clearly articulated (who, what when, where, why) to allow for reproducibility.<br>2. Ethical, explicit and detailed methods sections for recruitment, data collection and analysis.<br>3. Comprehensive reporting of data and accurate interpretation clearly warranted from findings.<br>Medium:<br>1.Intervention described but not with sufficient detail to replicate.<br>2.Satisfactory reporting of methods sections for ethics, recruitment, data collection and analysis—minor omissions evident.<br>3.Minor bias in reporting of data and interpretations partially warranted from findings.<br>Low:<br>1. Intervention not described.<br>2. Unsatisfactory methods sections for data collection and analysis with major omissions evident.<br>3. Unsatisfactory reporting of data and/or interpretations not warranted from findings. | High:<br>Large-scale multimethod designs for example, Randomised Controlled Trials, quasi-experimental using validated tools and/or interview, observational data with large sample size.<br>Medium:<br>Medium-scale qualitative, quantitative or mixed-method designs, for example, local pre–post, postsurvey/questionnaire using acceptable tools and/or interview, observational data with medium/small sample size.<br>Low: Study design unclear or inappropriate choice of tools used. | High:<br>Provides useful and relevant answers closely relatable to review objectives.<br>Medium:<br>Provides answers relatable to review objectives.<br>Low: Unable to provide answers relatable to the review objectives. | High:<br>A must be accredited no less than a 'high' in two domains to achieve an overall WoE of high.<br>Medium:<br>A must be accredited no less than a 'medium' in two domains to achieve an overall WoE of Medium.<br>Low: A must be accredited a 'low' in two domains to be excluded. |

EPPI, Evidence for Policy and Practice Information and Co-ordinating Centre.

field of QI education and continue knowledge development around APPs working.

## ETHICS AND DISSEMINATION

No ethical approval was required to undertake this study. This realist review is the first to our knowledge that aims to develop an understanding around how, and under what contexts, APPs collaborate to implement QI education in preregistration nursing curriculums. This study is aimed at international nurse educators and nurse leaders who can make sense of the findings to adapt their implementation approach within the limits of their own education and practice contexts. We will disseminate findings through conference presentations and peer-reviewed journals.

**Contributors** LA, CM and PT made substantial contributions to the conception and design of the protocol development. All authors have approved the submitted version and have agreed to be personally accountable for own contributions and ensure questions related to the accuracy or integrity of the work, even ones in which the author was not personally involved, are appropriately investigated, resolved and the resolution documented in the literature.

**Funding** The authors have not declared a specific grant for this research from any funding agency in the public, commercial or not-for-profit sectors.

**Competing interests** None declared.

**Patient and public involvement** Patients and/or the public were not involved in the design, or conduct, or reporting, or dissemination plans of this research.

**Patient consent for publication** Not applicable.

**Provenance and peer review** Not commissioned; externally peer reviewed.

**ORCID iD**
Lorraine Armstrong http://orcid.org/0000-0001-8951-2712

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
