## [Reviewer comments · BMJ Open]

ARTICLE DETAILS

TITLE (PROVISIONAL)	How, and under what contexts, do academic-practice partnerships collaborate to implement healthcare improvement education into preregistration nursing curriculums: a realist review protocol.
AUTHORS	Armstrong , Lorraine; Moir, Chris; Taylor, Peta

VERSION 1 – REVIEW

REVIEWER	Paul , Lorena University of the Incarnate Word Ila Faye
REVIEW RETURNED	25-Aug-2023

GENERAL COMMENTS	The study protocol incorporates a clearly stated research question and the intended study outcomes (i.e., aim and objectives) on page 10. The research question is presented in the abstract and further expanded on page 6. The proposed realist review method and study design are appropriate for answering such questions related to circumstantial contexts (Gerstein Science Information Centre, 2023). The proposed application of RAMSES guidelines is appropriate and indicated for the realist review process (Wong, et al., 2016) and for answering the proposed research questions. The Application of Evidence for Policy and Practice Information (EPPI, 2010) framework is commonly applied to the realist review method and will facilitate the accomplishment of informative article appraisals. It was noted that some references are dated greater than 10 years, but these documents are informative in terms of historical and strategic national health care system visions, perspectives, and guidance. Therefore, I would consider these older dated articles relevant to the proposed iterative discovery of context themes. The authors clearly acknowledge potential limitations of the study on page 3. These align with Gerstein Science Information Centre's (2023) position that realist review studies are generally limited in terms of reproducibility and strength of recommendations. References: Evidence for Policy and Practice Information and Co-ordinating Centre. (2010). EPPI-centre methods for conducting systemic reviews. Gerstein Science Information Centre. (2023). Knowledge syntheses: Systematic & scoping reviews, and other Review Types. https://guides.library.utoronto.ca/c.php?g=713309&p=5105450 Wong, G., Westhorp, G., Manzano, A., Greenhalgh, J., Jagosh, J., & Greenhalgh, T. (2016). RAMESES II reporting standards for realist evaluations. BMC Medicine, 14(1), 1-18.
--

REVIEWER	Woodward, Kyla F. University of Washington, Child, Family, and Population Health Nursing
-----------------	---

REVIEW RETURNED	29-Aug-2023
GENERAL COMMENTS	The review protocol is clear, straightforward, and well justified. The authors state an appropriate focus on context as a critical element of understanding intervention. I appreciate the thoughtful identification and application of theory underlying the study.

VERSION 1 – AUTHOR RESPONSE

2. Reviewer: 1 Comments to Author

a) Thank you for reviewing and applying your expertise to the realist review protocol and for considering that our methods and references are appropriate to answer our proposed research questions.

3. Reviewer: 2 Comments to Author

a) Thank you for reviewing and applying your expertise to the realist review protocol for your encouragement in respect of our theoretical approach to answer our proposed research questions.